# Genetic evidence that Nkx2.2 acts primarily downstream of Neurog3 in pancreatic endocrine lineage development

Angela J Churchill[1,2,3], Giselle Dominguez Gutiérrez[1,2,4], Ruth A Singer[1,2,5], David S Lorberbaum[6], Kevin A Fischer[6], Lori Sussel[1,2,3,4,5,6]*

[1]Naomi Berrie Diabetes Institute, Columbia University Medical School, New York, Columbia; [2]Department of Genetics and Development, Columbia University Medical School, New York, Columbia; [3]Genetics and Development Doctoral Program, Columbia University Medical School, New York, Columbia; [4]Nutritional and Metabolic Biology Doctoral Program, Columbia University Medical School, New York, Columbia; [5]The Integrated Graduate Program in Cellular, Molecular and Biomedical Studies, Columbia University Medical School, New York, Columbia; [6]Barbara Davis Center, University of Colorado, Denver, United States

**Abstract** Many pancreatic transcription factors that are essential for islet cell differentiation have been well characterized; however, because they are often expressed in several different cell populations, their functional hierarchy remains unclear. To parse out the spatiotemporal regulation of islet cell differentiation, we used a *Neurog3-Cre* allele to ablate *Nkx2.2*, one of the earliest and most broadly expressed islet transcription factors, specifically in the Neurog3[+] endocrine progenitor lineage (*Nkx2.2^{△endo}*). Remarkably, many essential components of the $\beta$ cell transcriptional network that were down-regulated in the *Nkx2.2^{KO}* mice, were maintained in the *Nkx2.2^{△endo}* mice - yet the *Nkx2.2^{△endo}* mice displayed defective $\beta$ cell differentiation and recapitulated the *Nkx2.2^{KO}* phenotype. This suggests that Nkx2.2 is not only required in the early pancreatic progenitors, but has additional essential activities within the endocrine progenitor population. Consistently, we demonstrate Nkx2.2 functions as an integral component of a modular regulatory program to correctly specify pancreatic islet cell fates.

*For correspondence: lori.sussel@ucdenver.edu

Competing interests: The authors declare that no competing interests exist.

## Introduction

The pancreas is an essential dual-functioning organ that contains an exocrine and endocrine compartment. The exocrine pancreas represents the majority of the pancreatic tissue and contains acinar cells that secrete digestive enzymes and the ductal cells through which the enzymes are transported into the duodenum. The endocrine pancreas only represents 1–2% of the organ, but is comprised of hormone-producing islet cells that are critical for the maintenance of glucose homeostasis. Interestingly, although these two compartments have highly disparate functions, they arise from a common multipotent progenitor population that gradually forms spatially and molecularly distinct progenitor subpopulations during development (*Burlison et al., 2008*; *Offield et al., 1996*; *Pan and Wright, 2011*; *Zhou et al., 2007*). A bi-potent duct/endocrine progenitor population is then derived from the multipotent progenitor population and subsequently differentiates into a highly restricted endocrine progenitor population that produces all islet cell types, including the functionally important insulin-producing $\beta$ cell lineage.

The progressive restriction of progenitor-cell lineage potential to ultimately give rise to pancreatic islet cells is believed to be regulated by the sequential expression of several essential transcription factors. The multipotent progenitor population is defined by the expression of Pancreatic and duodenal homeobox 1 (Pdx1) and Pancreas transcription factor 1a (Ptf1a), transcriptional regulators that are essential for pancreatic formation; deletion of *Pdx1* or *Ptf1a* leads to pancreatic agenesis (*Burlison et al., 2008*; *Gu et al., 2003*; *Jonsson et al., 1994*; *Kawaguchi et al., 2002*; *Offield et al., 1996*). A subset of cells from the Pdx1⁺ Ptf1a⁺ multipotent progenitors gradually become restricted to the endocrine lineage with the expression of Neurogenin 3 (Neurog3), which delineates the endocrine progenitor population. Neurog3 is essential for the formation of all six embryonic pancreatic endocrine cell types: the insulin-producing β cells, glucagon-producing α cells, somatostatin-producing δ cells, pancreatic polypeptide-producing PP cells, gastrin-producing G cells, and ghrelin-producing ε cells (*Gradwohl et al., 2000*; *Gu et al., 2002*; *Heller et al., 2005*; *Schwitzgebel et al., 2000*; *Suissa et al., 2013*). Mice lacking *Neurog3* almost completely fail to form these endocrine lineages (*Gradwohl et al., 2000*; *Heller et al., 2005*; *Suissa et al., 2013*).

Although it is well-established that β cells are derived from the Neurog3⁺ progenitor lineage, there is still some uncertainty about when and where the β cell lineage is specified during pancreas development. β cell fate may be influenced early within the Pdx1⁺ pancreatic progenitor to specify unipotent populations of Neurog3⁺ endocrine progenitor cells (*Desgraz and Herrera, 2009*) or it may be occurring within the Neurog3⁺ endocrine progenitor lineage after induction of *Neurog3*. Interestingly, *Johansson et al. (2007)* demonstrated that pancreatic progenitor cells transition through distinct competence windows in response to *Neurog3* induction for each endocrine cell fate, with β cell competence being acquired during a window of development between embryonic day (E)10.5 and E16 (*Johansson et al., 2007*).

Many of the transcription factors that are essential for β cell specification have been identified, including Nkx2.2, Nkx6.1, Rfx6, Glis3, Insm1 and Neurod1; however, we still have an incomplete understanding of how these factors regulate the timing and mechanism of β cell fate induction since they are often expressed both early and throughout pancreas development and are not always restricted to the β cell lineage. For example, Nkx2.2, Nkx6.1 and Rfx6 are expressed in multipotent pancreatic progenitors *and* endocrine progenitors, in addition to their maintained expression in several of the mature endocrine lineages (*Arnes et al., 2012b*; *Jørgensen et al., 2007*; *Pedersen et al., 2005*; *Soyer et al., 2010*). Glis3 expression is first detected in the bipotent progenitor cells, but is then maintained in the preductal and endocrine progenitors, and ultimately restricted to the β, PP and ductal lineages (*Kang et al., 2016*). Insm1 is similarly expressed in the pancreatic pre-endocrine cells, endocrine progenitor cells and is maintained in all endocrine cell types (*Gierl et al., 2006*; *Mellitzer et al., 2006*; *Osipovich et al., 2014*). Lastly, *Neurod1* can be detected as early as E9.5 in the early glucagon and ghrelin producing cells, but predominantly becomes restricted first to endocrine progenitors and then is maintained in all endocrine lineages (*Anderson et al., 2009a*; *Naya et al., 1997*). Null mutations of all six of these transcriptional regulators cause neonatal lethality likely due to their severe defects in β cell development (*Gierl et al., 2006*; *Naya et al., 1997*; *Osipovich et al., 2014*; *Prado et al., 2004*; *Smith et al., 2010*; *Sussel et al., 1998*); however, it is unknown exactly when each of these factors exerts their functional role in β cell fate specification.

To gain a better understanding of both the timing and mechanism of β cell fate induction we chose to focus on the essential islet transcription factor, Nkx2.2. Nkx2.2 is a critical regulator of appropriate islet cell lineage specification during pancreagenesis; mice carrying an *Nkx2.2* null mutation form reduced numbers of α and β cells and instead form increased numbers of ghrelin-producing ε cells due to defects in cell lineage specification (*Prado et al., 2004*; *Sussel et al., 1998*). Importantly, β cell fate specification is completely dependent on Nkx2.2 as no β cells are formed in mice carrying a null mutation in *Nkx2.2* (*Nkx2.2*ᴷᴼ) (*Sussel et al., 1998*). Nkx2.2 is co-expressed with Pdx1 throughout the pancreatic progenitor domain, but its expression becomes restricted to the Neurog3⁺ endocrine progenitor population (*Arnes et al., 2012b*; *Jørgensen et al., 2007*; *Sussel et al., 1998*). Nkx2.2 also appears to function at or near the top of the regulatory cascade of transcription factors to specify β cell fate: The expression of *Neurog3*, *Nkx6.1*, *Rfx6* and *Neurod1* was significantly reduced in *Nkx2.2*ᴷᴼ progenitor populations (*Anderson et al., 2009a*, *2009b*; *Chao et al., 2007*) (*Table 1*), suggesting that Nkx2.2 functions within the pancreatic

**Table 1.** $NKx2.2^{\triangle endo}$ and $Nkx2.2^{KO}$ E15.5 RNA-Seq - Gene expression changes in select transcription factors and pancreatic hormones.

| | $Nkx2.2^{\triangle endo}$ fold change | Adjusted p-value | $Nkx2.2^{KO}$ fold change | Adjusted p-value |
|---|---|---|---|---|
| **Transcription factors** | | | | |
| Arx | 2.16 | 3.26E-09 | 1.57 | 0.0031 |
| MafA | 0.11 | 3.39E-06 | 0.13 | 0.0040 |
| MafB | 0.43 | 1.31E-10 | 0.22 | 3.92E-38 |
| NeuroD1 | 0.89 | 1 | 0.32 | 3.01E-23 |
| Neurog3 | 1.11 | 1 | 0.59 | 0.0198 |
| Nkx6.1 | 0.77 | 1 | 0.82 | 0.9684 |
| Pax4 | 1.02 | 1 | 0.73 | 0.4762 |
| Pax6 | 0.36 | 1.07E-06 | 0.21 | 2.84E-30 |
| Pdx1 | 0.88 | 1 | 1.25 | 0.7916 |
| Rfx6 | 1.00 | 1 | 0.63 | 0.0013 |
| Sox9 | 0.94 | 1 | 1.08 | 1 |
| **Pancreatic hormones** | | | | |
| Gast | 0.19 | 2.16E-07 | 0.09 | 2.63E-09 |
| Gcg | 0.16 | 2.23E-12 | 0.01 | 4.64E-16 |
| Ghrl | 8.27 | 1.94E-31 | 6.43 | 6.72E-16 |
| Ins1 | 0.03 | 4.84E-24 | 0.00 | 4.32E-11 |
| Ins2 | 0.01 | 2.99E-55 | 0.00 | 1.48E-11 |
| Ppy | 10.08 | 4.99E-42 | 2.17 | 7.15E-05 |
| Sst | 1.14 | 1 | 0.45 | 0.5130 |

epithelium to influence the derivation of endocrine-committed cells that can enter the appropriate endocrine lineage pathways.

The early expression of Nkx2.2 within the multipotent progenitor population, upstream of Neurog3 prompted us to determine whether Nkx2.2 also functioned downstream of Neurog3 to influence islet cell specification. We used *Neurog3-Cre; Nkx2.2^{flox/flox}* (*Nkx2.2^{\triangle endo}*) mice to ablate Nkx2.2 specifically within the Neurog3$^+$ endocrine progenitor lineage. Although Nkx2.2 is expressed upstream of Neurog3 and regulates *Neurog3* expression (*Anderson et al., 2009b*; *Sussel et al., 1998*), deletion of *Nkx2.2* using precise Neurog3-based floxed gene inactivation resulted in severe endocrine defects similar to the *Nkx2.2^{KO}* phenotype. Notably, unlike the *Nkx2.2^{KO}* mice, *Neurog3* expression in the *Nkx2.2^{\triangle endo}* mice was not affected. This suggests that Nkx2.2-mediated endocrine specification is established within the endocrine lineage, downstream of *Neurog3* activation. Furthermore, unlike the *Nkx2.2^{KO}* mice, expression of several essential islet cell transcription factors, including *Rfx6* and *Neurod1*, was unaffected. Co-occupancy analysis revealed significant overlap between Nkx2.2 direct targets and Rfx6 and Neurod1 bound genes. This indicates $\beta$ cell formation may not be regulated by a simple linear transcriptional cascade that functions downstream of Nkx2.2, and implicates Nkx2.2 as an essential component of a modular $\beta$ cell transcription network.

## Results

### *Nkx2.2* is efficiently ablated in *Nkx2.2^{\triangle endo}* mice

Nkx2.2 is expressed at the onset of pancreas development throughout the multipotent progenitor domain, but subsequently becomes restricted to the Neurog3$^+$ endocrine progenitor population (*Jørgensen et al., 2007*). Furthermore, null mutations in *Nkx2.2* (*Nkx2.2^{KO}*) lead to significantly reduced *Neurog3* RNA expression, although there is no corresponding change in Neurog3$^+$ cell numbers (*Anderson et al., 2009b*). To determine whether Nkx2.2 activity downstream of Neurog3

was also necessary for specific aspects of endocrine cell specification, *Neurog3-Cre; Nkx2.2^flox/flox* or *Nkx2.2^flox/LacZ; R26R-Tomato* (hereafter referred to as *Nkx2.2^△endo*) mice were generated to ablate *Nkx2.2* specifically within the endocrine progenitor cells and their descendants (*Madisen et al., 2010*; *Mastracci et al., 2013*; *Schonhoff et al., 2004*). Efficient deletion of *Nkx2.2* in the *Nkx2.2^△endo* mice was demonstrated by a reduction in *Nkx2.2* mRNA expression at E15.5 and E18.5 (*Figure 1A*). Residual *Nkx2.2* gene expression at E15.5 likely reflects the *Nkx2.2* expression in pancreatic progenitor cells that are still present at this stage of development. Consistent with the decrease in *Nkx2.2* gene expression, Nkx2.2 protein was almost undetectable in the pancreas at E15.5 and postnatal day (P)0 (*Figure 1B,C*, *Figure 1—figure supplement 1*). Activation of the *R26R-Tomato* allele was also used to monitor accurate Cre activity within the *Nkx2.2^△endo* mice and, accordingly, Tomato+ (Tom+) cells were found specifically within the endocrine compartment (*Figure 1B,C*, *Figure 1—figure supplement 1*).

Deletion of *Nkx2.2* from the endocrine progenitor population did not affect the number of Neurog3-expressing endocrine progenitor cells and Neurog3 expression within the Tomato-labeled, Nkx2.2-deficient cells was maintained (*Figure 1D,E*, *Figure 1—figure supplement 1*, *Table 1*).

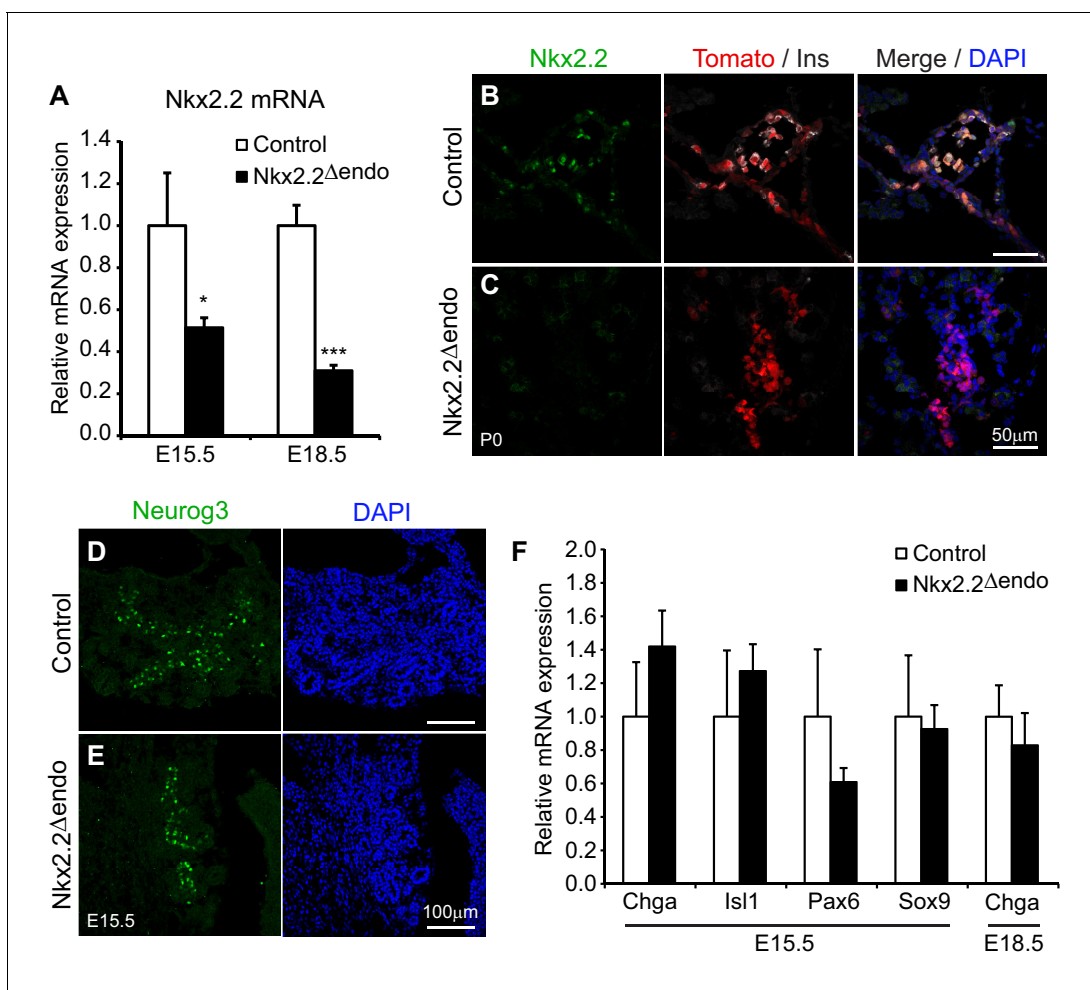

**Figure 1.** Nkx2.2 is ablated specifically in the endocrine lineage in *Nkx2.2^△endo* mice. (**A**) Nkx2.2 gene expression is decreased at E15.5 (Control n = 3, *Nkx2.2^△endo* n = 6) and E18.5 (Control n = 6, *Nkx2.2^△endo* n = 4). (**B–C**) Immunostaining for Nkx2.2 shows decreased expression in *Nkx2.2^△endo* mice at P0 (n = 3). (**D–E**) Immunostaining for Neurog3 shows no apparent changes in the Neurog3+ cell population at E15.5 (n = 3). (**F**) Endocrine compartment gene expression is unchanged at E15.5 (Control n = 3, *Nkx2.2^△endo* n = 6) and E18.5 (Control n = 6, *Nkx2.2^△endo* n = 4). (*) p<0.05; (***) p<0.001.

The following figure supplement is available for figure 1:

**Figure supplement 1.** Exocrine and endocrine compartment size is unaffected *Nkx2.2^△endo* mice.

Deletion of Nkx2.2 did not appear to affect the earlier cell populations that give rise to endocrine progenitors since the expression of *Sox9*, an early progenitor marker that is expressed immediately upstream of *Neurog3* in the bipotent progenitor pool from which endocrine cells are derived, was unaffected (*Figure 1F*). Consistent with specific deletion of Nkx2.2 from the endocrine progenitor population, there were also no obvious changes in the exocrine lineage, as indicated by amylase expression (*Figure 1—figure supplement 1*). Expression of pan-endocrine genes, such as *ChromograninA* (*Chga*), *Isl1*, and *Pax6* was also unchanged in the absence of *Nkx2.2* at E15.5, and the area and number of Tomato-labeled endocrine lineage cells was also unaffected in the *Nkx2.2*$^{\triangle endo}$ mice at E15.5 and P0, suggesting endocrine compartment size was not decreased when Nkx2.2 was deleted from the endocrine progenitor cell (*Figure 1B,C,F*, *Figure 1—figure supplement 1*).

## *Nkx2.2*$^{\triangle endo}$ mice display similar endocrine defects as *Nkx2.2*$^{KO}$ mice

Since Nkx2.2 is necessary for the establishment of proper endocrine cell ratios, we assessed the status of endocrine cell differentiation in the *Nkx2.2*$^{\triangle endo}$ mice during the course of pancreas development. Similar to the *Nkx2.2*$^{KO}$ mice (*Prado et al., 2004*; *Sussel et al., 1998*), *Nkx2.2*$^{\triangle endo}$ mice displayed a reduction of insulin$^+$ (Ins$^+$) cells and glucagon$^+$ (Gcg$^+$) cells, and increased ghrelin$^+$ (Ghrl$^+$) cells at E15.5 and P0 (*Figure 2A–H*; *Figure 2—figure supplement 1*). Gastrin$^+$ (Gast$^+$) cells were decreased at E15.5 (*Figure 2I,J*), and no changes were observed in somatostatin$^+$ (SST$^+$) cell number at P0 (*Figure 2K–L*; *Figure 2—figure supplement 1*). Unlike the Nkx2.2$^{KO}$ mice which lack all β cells, a small number of Ins$^+$ cells were detected (*Figure 2A–L*; *Figure 2—figure supplement 1*), which is possibly due to the inability of the *Neurog3-Cre* allele to completely inactivate *Nkx2.2*. Gene expression at E15.5 and E18.5 corresponded to changes in endocrine cell numbers. *Insulin* (*Ins1* and *Ins2*) and *glucagon* expression were significantly decreased; there was increased *ghrelin* expression, and no change in *somatostatin* gene expression at E15.5 and E18.5 (*Figure 2M*). *Gast* gene expression was also decreased at E15.5, but *Gast* expression was extinguished from both the wild type and mutant pancreata by E18.5 (*Figure 2M*), as previously observed (*Suissa et al., 2013*). Similar to the *Nkx2.2*$^{KO}$ mice, the *Nkx2.2*$^{\triangle endo}$ mice die shortly after birth between P2 and P4 (*Figure 2N*).

Interestingly, we observed increased *pancreatic polypeptide* (*Ppy*) expression at E15.5 and E18.5 in *Nkx2.2*$^{\triangle endo}$ embryos (*Figure 2M*). Although this was in contrast to previous studies in which we had observed decreased *Ppy* gene expression and PP$^+$ cell numbers (*Chao et al., 2007*; *Prado et al., 2004*; *Sussel et al., 1998*), we confirmed that there was similar upregulation of *Ppy* in the *Nkx2.2*$^{LacZ/LacZ}$ (*Nkx2.2*$^{KO}$) pancreas (*Arnes et al., 2012b*) compared to the *Nkx2.2*$^{\triangle endo}$ pancreas, although the magnitude of upregulation was greater in the *Nkx2.2*$^{\triangle endo}$ embryos (*Figure 2M, O*). In *Nkx2.2*$^{\triangle endo}$ mice, upregulation of *Ppy* RNA correlated with changes in protein; immunostaining revealed increased PP$^+$ cells at both E15.5 and P0 (*Figure 2—figure supplement 2*). While some Ins$^+$PP$^+$ cells could be observed (arrow in *Figure 2—figure supplement 2*), there was no significant difference between *Nkx2.2*$^{\triangle endo}$ mice and controls (*Figure 2—figure supplement 2*), suggesting that similar to the ghrelin-expressing cells, whose increased numbers were due to altered ε cell specification at the expense of β cell specification, the excess PP cells did not arise from misexpression of PP in β cells or as a result of β cell reprogramming. We speculate that the novel *Ppy* phenotype in the *Nkx2.2*$^{\triangle endo}$ and *Nkx2.2*$^{KO}$ mice may result from a change in mouse strain background (see Discussion).

## *Nkx2.2*$^{\triangle endo}$ and *Nkx2.2*$^{KO}$ mice share important gene expression changes

It was somewhat unexpected that the restricted deletion of *Nkx2.2* in the Neurog3$^+$ progenitor cells resulted in a phenotype that was remarkably similar to the *Nkx2.2*$^{KO}$ mice, which lacked *Nkx2.2* throughout pancreas development. To more precisely determine the similarities and differences between the *Nkx2.2*$^{\triangle endo}$ and *Nkx2.2*$^{KO}$ mice during the process of endocrine cell differentiation, we performed RNA-Seq analysis on whole pancreata derived from *Nkx2.2*$^{\triangle endo}$ embryos, *Nkx2.2*$^{KO}$ embryos, and their respective littermate controls at E15.5, a time of relatively high Neurog3$^+$ cell abundance. Consistent with their similar phenotypes, there was significant overlap (120 genes, p-value <2.2e-16, *Figure 3A*) between the differentially expressed genes (adjusted p-value <0.05), including genes encoding pancreatic hormones (*Figure 3B*) and important α and β cell regulatory

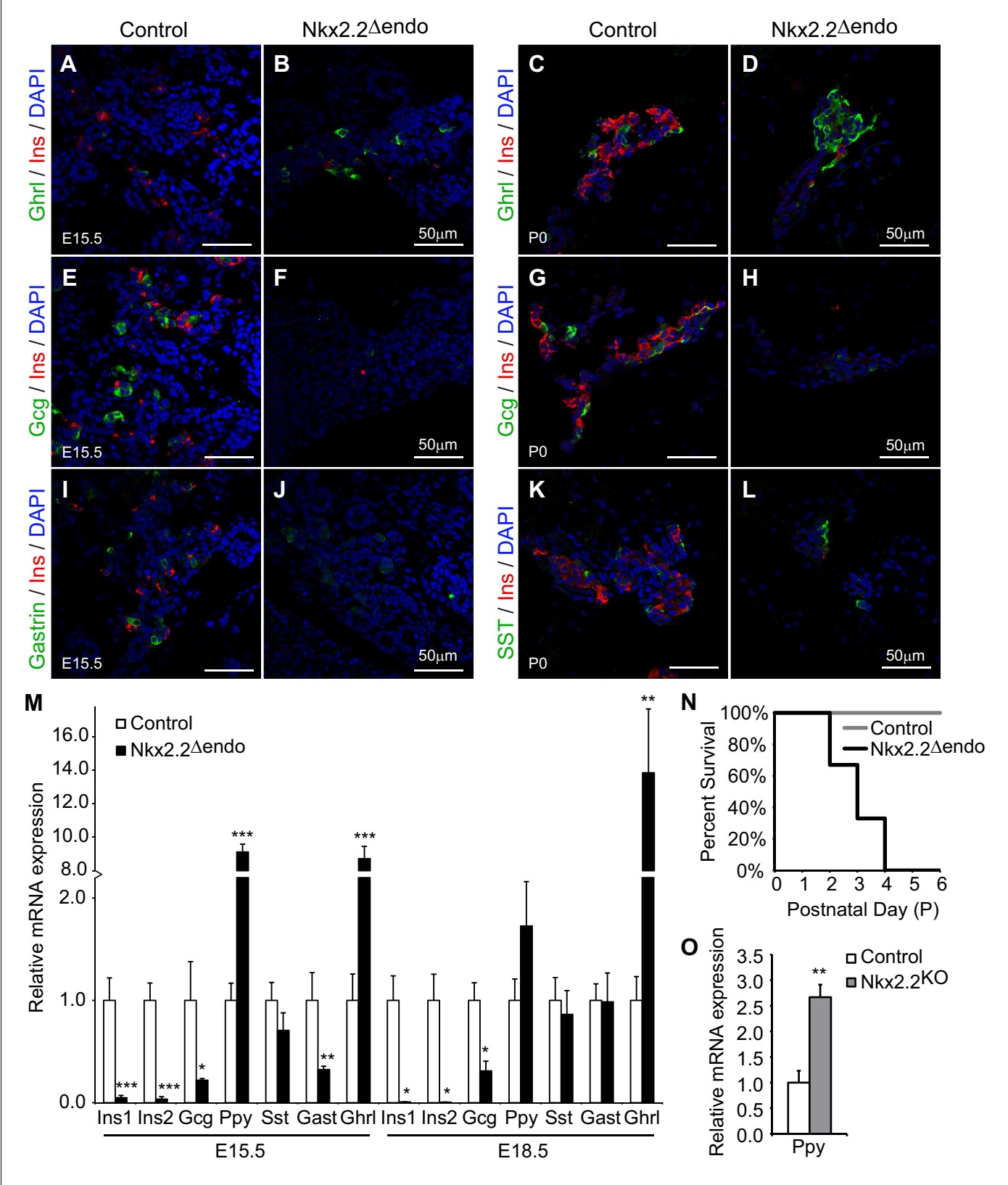

**Figure 2.** *Nkx2.2*$^{\triangle endo}$ mice recapitulate the *Nkx2.2*$^{KO}$ phenotype. (A–H) Ghrl$^+$ cells are increased, Ins$^+$ cells are greatly reduced, and Gcg$^+$ cells are decreased at E15.5 and P0 in *Nkx2.2*$^{\triangle endo}$ mice (n = 3). (I–J) Gast$^+$ cells are reduced in *Nkx2.2*$^{\triangle endo}$ embryos at E15.5 (n = 3). (K–L) SST$^+$ cells are
*Figure 2 continued on next page*

*Figure 2 continued*
unchanged in $Nkx2.2^{\triangle endo}$ mice at P0 (n = 3). (**M**) Hormone gene expression changes recapitulate $Nkx2.2^{KO}$ mice at E15.5 (Control n = 3, $Nkx2.2^{\triangle endo}$ n = 6) and E18.5 (Control n = 6, $Nkx2.2^{\triangle endo}$ n = 4). *Ppy* gene expression is increased at these time points. (**N**) $Nkx2.2^{\triangle endo}$ mice die neonatally. (**O**) qRT-PCR confirms increased *Ppy* at E15.5 in $Nkx2.2^{KO}$ embryos (Control n = 3, $Nkx2.2^{KO}$ n = 3). (*) p<0.05; (**) p<0.01; (***) p<0.001.
The following figure supplements are available for figure 2:

**Figure supplement 1.** $Nkx2.2^{\triangle endo}$ mice display endocrine specification defects at P0.
**Figure supplement 2.** PP cells are increased in $Nkx2.2^{\triangle endo}$ mice.

factors, such as *Mafa* and *Mafb* (*Figure 3C,D*; *Table 1*). Furthermore, several transcription factor genes expressed in endocrine progenitors and involved in β cell specification, such as *Nkx6.1*, *Pax4*, and *Pdx1*, that were unchanged $Nkx2.2^{KO}$ embryos at E15.5 were also not affected in $Nkx2.2^{\triangle endo}$ pancreas at this early time point (*Figure 3—figure supplement 1*; *Table 1*).

## Key islet regulators Neurog3, Neurod1, and Rfx6 are retained in $Nkx2.2^{\triangle endo}$ mice

Notably, in sharp contrast to the $Nkx2.2^{KO}$ mice, which have reduced *Neurog3*, *Neurod1*, and *Rfx6* (*Anderson et al., 2009a*, *2009b*; *Chao et al., 2007*) (*Table 1*), expression of *Neurog3*, *Neurod1*, and *Rfx6* were unaffected in the $Nkx2.2^{\triangle endo}$ mice (*Figure 3E*). We performed qRT-PCR analysis to confirm the maintenance of *Neurog3*, *Neurod1*, and *Rfx6* gene expression in E15.5 and E18.5 $Nkx2.2^{\triangle endo}$ mice (*Figure 3F*). Although *Neurog3* mRNA appeared to be modestly increased at E18.5 in the $Nkx2.2^{\triangle endo}$ mice (*Figure 3F*), Neurog3+ cells could not be detected by immunostaining in perinatal animals (*Figure 3G,H*). These results suggest that although Nkx2.2 regulates *Neurog3*, *Neurod1*, and *Rfx6* expression within the pancreatic progenitor population, regulation of *Neurog3*, *Neurod1*, and *Rfx6* expression within the endocrine progenitor cells becomes independent of Nkx2.2. Furthermore, this suggests that maintenance of *Neurog3*, *Neurod1*, and *Rfx6* expression in the absence of *Nkx2.2* is not sufficient for appropriate islet lineage specification.

## Nkx2.2 direct target genes are simultaneously bound by Rfx6 and/or Neurod1

The finding that β cell specification was disrupted, even though *Neurog3*, *Neurod1* and *Rfx6* expression remained intact, could indicate that Nkx2.2 functions as an obligate co-factor for these proteins in the regulation of β cell specification. This is supported by our previous observation that Nkx2.2 cooperated with Neurog3 to regulate the expression of Neurod1 (*Anderson et al., 2009a*). To determine whether there was a general requirement for a cooperative role for Nkx2.2 within the known β cell specification regulatory pathways, we assessed whether Nkx2.2 co-regulated the β cell program with Rfx6 and Neurod1. Nkx2.2 bound and regulated genes were identified by cross-referencing the differentially expressed genes in the $Nkx2.2^{\triangle endo}$ RNA-Seq (adjusted p-value <0.05, *Figure 3*) with an Nkx2.2 ChIP-Seq performed on the MIN6 β cell line (GSE79725) (*Gutiérrez et al., 2017*). Notably, the majority of the genes regulated by Nkx2.2 appeared to be primary targets: 169/175 of the significantly regulated genes were bound by Nkx2.2 within 100 kb of of their respective transcriptional start sites (TSS) (*Figure 3A*; *Figure 4A,C*). We then compared the Nkx2.2 direct targets to genes bound by either Rfx6 (GSE62844; 9406 genes) or Neurod1 (GSE30298; 5003 genes) (*Piccand et al., 2014*) (*Tennant et al., 2013*). This analysis revealed significant overlap of Rfx6 binding near Nkx2.2 target genes (120 genes, p-value <2.2e-16, *Figure 4A*) and gene ontology (GO) analysis indicated significant representation of genes related to hormone secretion (*Figure 4B*). Analysis of Neurod1 bound genes revealed similar overlap with Nkx2.2 targets (92 genes, p-value <2.2e-16, *Figure 4C*), and GO analysis identified significant representation of insulin secretion genes (*Figure 4D*). Furthermore, greater than 75% of the 92 genes simultaneously bound by Nkx2.2 and Neurod1 in the same or nearby regulatory regions were also bound by Rfx6 (72 genes, *Figure 4E*), demonstrating novel regulatory co-operation between these three transcription factors.

Although it is possible that at some genomic loci the Nkx2.2/Neurod1/Rfx6 co-regulation of important β cell genes could be due to direct protein-protein interactions, the respective binding

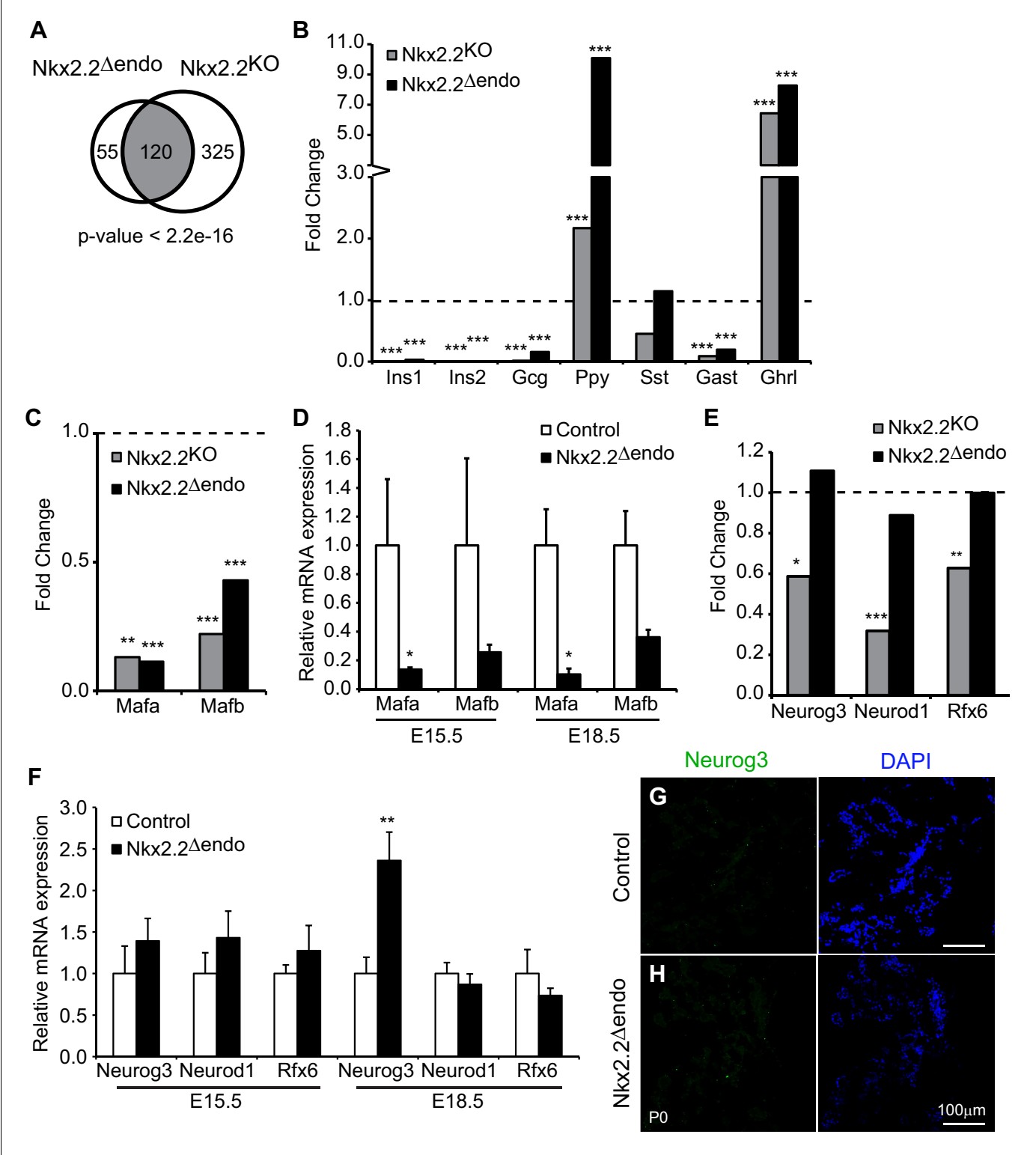

**Figure 3.** *Nkx2.2^△endo* and *Nkx2.2^KO* embryos have shared and distinct gene expression changes. (**A**) Comparison of *Nkx2.2^△endo* and *Nkx2.2^KO* RNA-Seq experiments shows significant overlap of dysregulated genes (adjusted p-value <0.05.) (**B**) RNA-Seq reveals similar changes in hormone expression in *Nkx2.2^△endo* and *Nkx2.2^KO* embryos, relative to littermate controls. (**C**) RNA-Seq shows decreases in *Mafa* and *Mafb* gene expression in *Nkx2.2^△endo* and *Nkx2.2^KO* embryos, relative to littermate controls. (**D**) qRT-PCR confirms decreases in *Mafa* and *Mafb* at E15.5 (Control n = 3, *Nkx2.2^△endo* n = 6) and E18.5 (Control n = 6, *Nkx2.2^△endo* n = 4). (**E**) RNA-Seq shows unchanged *Neurog3*, *Neurod1*, and *Rfx6* gene expression in *Nkx2.2^△endo* embryos; Nkx2.2^KO embryos show downregulation, relative to littermate controls. (**F**) qRT-PCR confirms RNA-Seq results. *Neurod1* and *Rfx6* gene expression are

*Figure 3 continued on next page*

*Figure 3 continued*

unchanged at E15.5 and E18.5. *Neurog3* gene expression is unchanged at E15.5 and increased at E18.5. (E15.5: Control n = 3, *Nkx2.2*$^{\triangle endo}$ n = 3–6; E18.5: Control n = 5–6, *Nkx2.2*$^{\triangle endo}$ n = 4). (G–H) Neurog3$^+$ cells are not detected at P0 (n = 3). Mutant RNA-Seq data was normalized to littermate controls; dotted lines refer to 1.0 fold change. (*) p<0.05; (**) p<0.01; (***) p<0.001.

The following source data and figure supplement are available for figure 3:

**Source data 1.** *Nkx2.2*$^{\triangle endo}$ and *Nkx2.2*$^{KO}$ E15.5 RNA-Seq: Genes showing significant differential expression in both *Nkx2.2*$^{\triangle endo}$ and *Nkx2.2*$^{KO}$ mice (120 genes).

**Source data 2.** *Nkx2.2*$^{\triangle endo}$ and *Nkx2.2*$^{KO}$ E15.5 RNA-Seq: Genes showing significant differential expression in only *Nkx2.2*$^{KO}$ mice (325 genes).

**Source data 3.** *Nkx2.2*$^{\triangle endo}$ and *Nkx2.2*$^{KO}$ E15.5 RNA-Seq: Genes showing significantly differential expression in only *Nkx2.2*$^{\triangle endo}$ mice (55 genes).

**Figure supplement 1.** Many islet transcription factor genes are relatively unchanged in Nkx2.2deltaendo embryos.

sites were often not closely clustered. Inspection of the Nkx2.2, Rfx6, and Neurod1 ChIP-Seq data revealed a variety of co-occupancy patterns in the regulatory elements associated with several candidate Nkx2.2-regulated genes (*Figure 4F*). For example, all three transcription factors showed overlapping binding peaks at the *Mlxipl* locus (*Figure 4G*). However, at a region upstream of *Mafa*, only Nkx2.2 and Neurod1 were present (*Figure 4H*); whereas Nkx2.2 and Rfx6 co-bound at the *Gast* locus (*Figure 4I*). In addition, Nkx2.2, Rfx6, and Neurod1 all bound near *G6pc2*; however, these peaks were present at distinct locations flanking the gene (*Figure 4J*). These results suggest that while Nkx2.2, Rfx6, and Neurod1 are individually critical for the regulation of the islet cell program, it is their cooperative function that may be required for the appropriate activation of a number of essential *β* cell genes.

## Discussion

The spatiotemporal progression of islet cell specification is not well understood. While many transcription factors that are essential for this process are well-characterized, their sustained expression throughout development and in several different cell types has made it challenging to identify the timing of their function. Nkx2.2 is expressed within the Pdx1$^+$ pancreatic multipotent progenitor population and its expression is maintained throughout pancreas development, gradually becoming restricted to the endocrine cell lineages (*Arnes et al., 2012b*; *Jørgensen et al., 2007*; *Sussel et al., 1998*). Global deletion of *Nkx2.2* causes the reduction of many essential regulatory factors, including *Neurog3*, *Neurod1* and *Rfx6* (*Anderson et al., 2009a*, *2009b*; *Chao et al., 2007*) (*Table 1*) and mis-specification of all endocrine lineages, except the somatostatin-expressing δ cells. The fact that these mice completely fail to form *β* cells, combined with the reduced expression of many *β* cell regulatory factors provided a strong argument for positioning Nkx2.2 at or near the top of the hierarchy for regulation of *β* cell specification. In this study, we ablated Nkx2.2 specifically within the endocrine progenitor lineage to determine whether Nkx2.2 also functions downstream of Neurog3 in regulating islet cell specification. Surprisingly, we found similar endocrine lineage changes in *Nkx2.2*$^{\triangle endo}$ mice compared to *Nkx2.2*$^{KO}$ mice, including the failure to form *β* cells (*Sussel et al., 1998*) (Schematized in *Figure 5*). However, unlike the *Nkx2.2*$^{KO}$ mice, expression of the essential *β* cell transcription factors *Neurog3*, *Rfx6*, and *Neurod1* was unaffected, suggesting that although Nkx2.2 regulates these factors in the pancreatic progenitor population, reduction of *Neurog3*, *Neurod1* and *Rfx6* expression is not an obligate step in the mis-specification of islet fates. This may indicate that although Neurog3, Neurod1 and Rfx6 are essential for specifying the *β* cell program, Nkx2.2 activity must be preserved to facilitate the functional integrity of the *β* cell regulatory pathway (*Figure 5*). Furthermore, co-occupancy analysis revealed significant overlap of Nkx2.2 direct targets with Rfx6 and Neurod1 binding, suggesting cooperative transcriptional regulation of these factors.

Given its early expression in the multipotent progenitor population, it was unexpected that deletion of Nkx2.2 from the Neurog3$^+$ cells was sufficient to recapitulate the *Nkx2.2*$^{KO}$ phenotype.

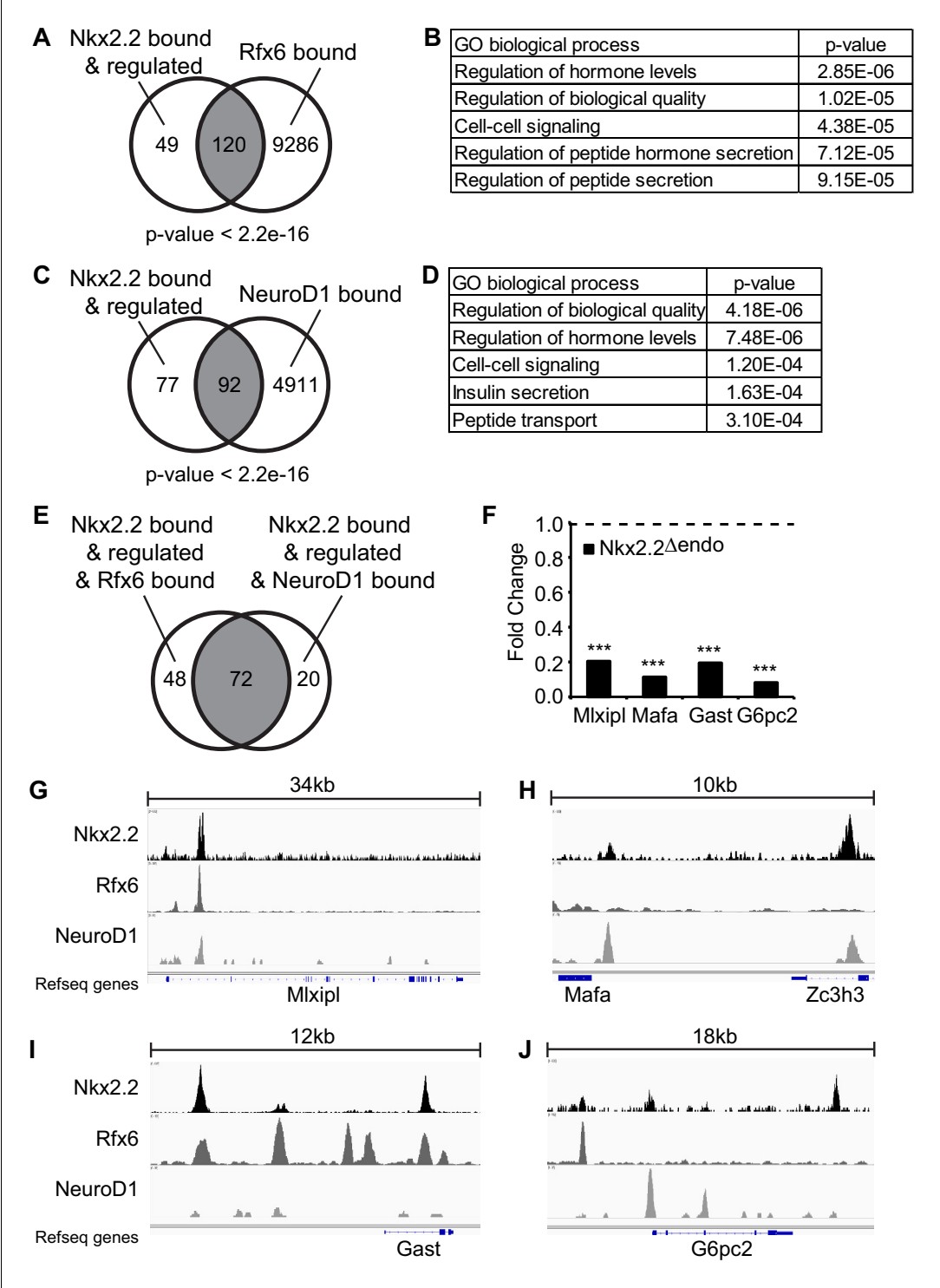

**Figure 4.** Nkx2.2 bound and regulated genes are co-bound by Rfx6 and Neurod1. (A) Comparison of Nkx2.2 bound and regulated genes and Rfx6 bound genes (B) Gene ontology (GO) analysis of genes with Nkx2.2 binding and regulation and Rfx6 binding. (C) Comparison of Nkx2.2 bound and regulated genes and Neurod1 bound genes. (D) GO analysis of genes with Nkx2.2 binding and regulation and Neurod1 binding. (E) Comparison between Nkx2.2 bound and regulated and Rfx6 bound genes with Nkx2.2 bound and regulated and Neurod1 bound genes. 72 genes are shared. (F) RNA-Seq shows decreased *Mlxipl*, *Mafa*, *Gast*, and *G6pc2* expression at E15.5 in *Nkx2.2^Δendo* embryos. Dotted line refers to 1.0 fold change. (***) p<0.001. (G) Nkx2.2, Rfx6, and Neurod1 show overlapping binding peaks at *Mlxipl* locus (mm9 chr5:135,580,783–135,615,792 shown). (H) Nkx2.2 and Neurod1 show overlapping binding peaks at a region upstream of *Mafa* (mm9 chr15:75,577,046–75,587,657 shown). (I) Nkx2.2 and Rfx6 show

*Figure 4 continued on next page*

*Figure 4 continued*

overlapping binding peaks upstream of *Gast* and within intron 1 (mm9 chr11:100,186,915–100,199,407 shown). (J) Non-overlapping Nkx2.2, Rfx6, and Neurod1 binding peaks near *G6pc2* locus (mm9 chr2:69,052,146–69,070,781 shown).
The following source data is available for figure 4:

**Source data 1.** Nkx2.2 bound and regulated genes that are also Rfx6 bound (120 genes) or NeuroD1 bound (92 genes).
**Source data 2.** Nkx2.2 bound and regulated genes that are also Rfx6 bound and NeuroD1 bound (72 genes).

However, it is consistent with a recent study showing that *Neurog3-Cre; Rfx6^flox/flox* mice display $\beta$ cell loss similar to the *Rfx6^null* mice (*Piccand et al., 2014*). Similarly, Nkx6.1 functions downstream of Neurog3, with *Neurog3-Cre; Nkx6.1^flox/flox* mice displaying defective $\beta$ cell differentiation (*Schaffer et al., 2013*). These studies provide evidence that the endocrine progenitor cell, downstream of Neurog3 induction, is the primary site of islet cell specification. However, it is still possible that preprogramming of this population does occur, but that the respective programs can be overwritten once the cell has formed. The importance of Nkx2.2 function downstream of Neurog3 expression is also consistent with the timing of NKX2.2 in human islets; unlike mice, NKX2.2 is only first detected after the appearance of NEUROG3 progenitors (*Jennings et al., 2013*). However, while Nkx2.2 function within the mouse Pdx1⁺ pancreatic progenitor was not sufficient to allow correct islet cell specification and $\beta$ cell formation, further studies would be required to determine whether Nkx2.2 has additional functions at this earlier stage of pancreas development.

Although deletion of Nkx2.2 in the endocrine progenitor population appears to grossly phenocopy the *Nkx2.2^KO* phenotype, the phenotypes were not as severe as observed in the *Nkx2.2^KO* mice. Although this may be due to the efficiency of Cre-mediated deletion of the *Nkx2.2* allele, it could also suggest that there are partially redundant functions of Nkx2.2 in the pancreas progenitor and endocrine progenitor populations. In addition, there were over 300 genes significantly altered in *Nkx2.2^KO* pancreas that were unchanged in the *Nkx2.2^△endo* mice (*Figure 3A*; Table 4). Again, this could be due to the more penetrant phenotype associated with the *Nkx2.2^KO* allele or could suggest that Nkx2.2 may have additional non-redundant functions upstream of Neurog3. Future studies in which we are able to conditionally re-express Nkx2.2 in the Neurog3 population in an *Nkx2.2^KO* background, similar to the Neurog3 replacement studies by *Johansson et al. (2007)*, would possibly reveal these functions. There were also a small number of genes whose expression only appeared to be changed in the *Nkx2.2^△endo* mice (*Figure 3A*, Table 3). This is more difficult to explain unless

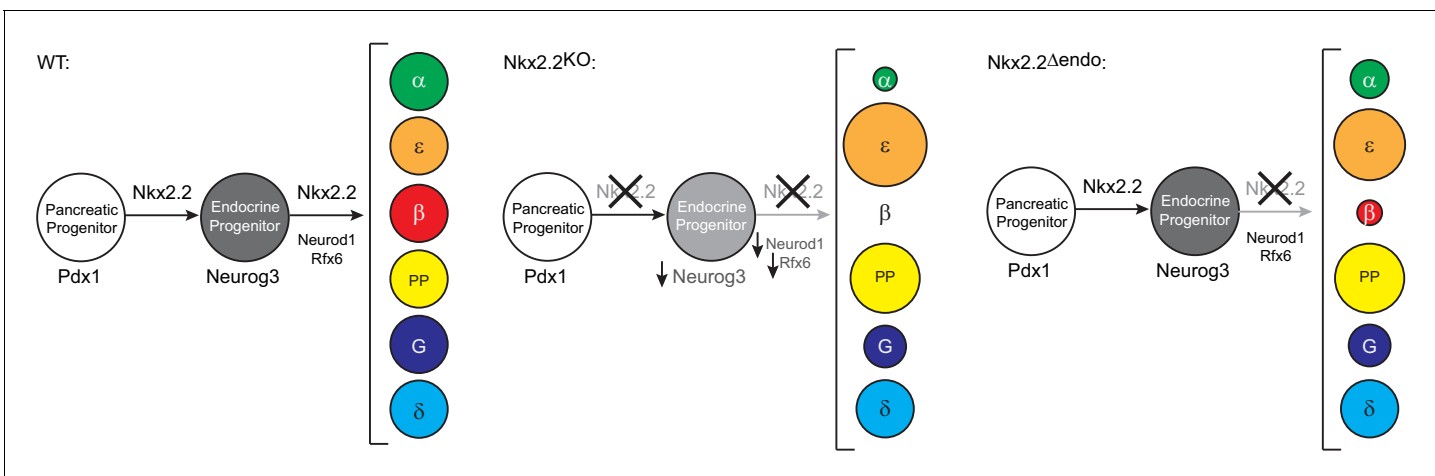

**Figure 5.** Model of transcriptional networks involved in endocrine cell specification in wildtype (WT), *Nkx2.2^KO*, and *Nkx2.2^△endo* conditions. Neurog3, Neurod1, and Rfx6 are not sufficient to allow for $\beta$ cell specification in the absence of Nkx2.2 in *Nkx2.2^△endo* mice. The size of the circles represents the proportion of cell numbers compared to wildtype.

there are some novel, compensatory gene changes that can occur in the remaining cell populations. Assessment of the identity of the differentially affected genes in either the $Nkx2.2^{KO}$ or $Nkx2.2^{\triangle endo}$ mice did not reveal any obvious mechanistic insights.

Several previous studies of the $Nkx2.2^{KO}$ mice demonstrated that Nkx2.2 is essential for the correct specification of all endocrine cell populations, except δ cells (*Arnes et al., 2012b*; *Mastracci et al., 2013*, *2011*; *Prado et al., 2004*; *Sussel et al., 1998*). These earlier studies also revealed that in the absence of Nkx2.2 there was mis-specification of the endocrine lineages, rather than reprogramming of the endocrine populations following their formation. Alternatively, deletion of the conserved tinman (TN) domain of Nkx2.2 resulted in both aberrant endocrine cell specification and reprogramming of β cells to the α cell fate (*Papizan et al., 2011*). Furthermore, recent studies in which $Nkx2.2$ was deleted from β cells after their specification resulted in β cell reprogramming into alternative endocrine lineages and the appearance of islet cells expressing more than one hormone (*Gutiérrez et al., 2017*). Interestingly, the phenotype of the $Nkx2.2^{\triangle endo}$ mice most closely resembles the $Nkx2.2^{KO}$ mice, and we do not see evidence of reprogramming or multi-hormonal expressing (transient or stable) endocrine cells in the mutant islets. However, since we are unable to genetically lineage trace the α or β cell populations in the $Nkx2.2^{\triangle endo}$ mice, we cannot completely rule out the possibility that reprogramming of the α or β cells contributes to the elevated number of ε or PP cells.

Comparison of Nkx2.2 direct targets with Rfx6 and Neurod1 binding revealed co-occupancy of these factors in the same or nearby enhancer elements located within proximity of an endocrine-related gene. While ChIP-Seq analyses can reveal novel insights into the transcriptional regulation of cell fate, these analyses were performed on chromatin isolated from the immortalized MIN6 β cell line in order to isolate sufficient material for the assay. It is important to note that although we only focused on potential gene targets that were dysregulated in the $Nkx2.2^{\triangle endo}$ pancreas, transcription factor binding in MIN6 cells may not accurately reflect binding in embryonic endocrine progenitor cells. Furthermore, our population-based transcriptome analysis is limited by the cell type heterogeneity present in the E15.5 pancreas, particularly when gene changes that are present in both progenitor populations and downstream sub-lineages may be represented. Future studies will benefit from single-cell RNA-Seq and ChIP-Seq analysis that will allow us to parse out the differential gene changes and transcription factor binding in the Nkx2.2-deficient endocrine progenitor cells.

We were surprised to observe an increase in the PP population of $Nkx2.2^{\triangle endo}$ mice. Since re-evaluation of *Ppy* expression in the $Nkx2.2^{KO}$ mice revealed a similar phenotype, the increase in PP is likely not due to a unique function of Nkx2.2 in the pancreatic progenitor to specify PP cells. The best explanation for the discrepant PP phenotype is that we moved the *Nkx2.2 null* allele to a C57Bl6/J background for this study. The upregulation of PP is also interesting since it parallels the increase we observed with ε cells. Consistent with the observation that a subset of the PP population is derived from Ghrl-expressing ε cells (*Arnes et al., 2012a*), PP$^+$Ghrl$^+$ cells can be infrequently found in $Nkx2.2^{\triangle endo}$ mice at P0 (data not shown). Furthermore, we observe an upregulation of *Arx* expression, which has also previously been shown to induce enhanced PP cell formation (*Collombat et al., 2007*). We are unable to determine whether these altered islet cell ratios are similarly due to β cell transdifferentiation events since the *Neurog3-Cre* labels all islet cell lineages.

Similar to many other developmental pathways, β cell specification involves the function of numerous transcription factors, some of which are expressed early and throughout tissue formation. Our results identify a novel critical window for Nkx2.2-mediated regulation of islet cell specification within the Neurog3 lineage following induction of Neurog3. Furthermore, although Nkx2.2 appears to function near the top of the islet transcriptional hierarchy, we demonstrate that maintained Nkx2.2 activity is also a necessary component of the regulatory networks functioning downstream of Neurog3 to induce β cell specification. Our discovery that different combinations of the set of transcription factors analyzed in this study directly regulate important genes in the β cell pathways raises important questions about the complex nature of cell-specific gene regulation that may have important implications for β cell induction, maturation and function. Furthermore, while previous research has focused on the initial expression of β cell factors, future studies may benefit by ensuring the prolonged and combinatorial expression of these factors to recapitulate in vivo development and allow cooperative induction of target genes.

# Materials and methods

## Animal maintenance

Mice were maintained on a C57BL/6J background (The Jackson Laboratory). *Neurog3-Cre* (RRID: IMSR:JAX:005667), *Nkx2.2^flox/flox or flox/LacZ* (RRID:MGI:5544100; RRID:BCBC_240), and *R26R-Tomato* (B6.Cg-*Gt(ROSA)26-Sor^tm14(CAG-tdTomato)Hze*/J; RRID:IMSR:RMRC13135) mice were genotyped as previously published (*Arnes et al., 2012b*; *Madisen et al., 2010*; *Mastracci et al., 2013*; *Schonhoff et al., 2004*). *Nkx2.2^LacZ/LacZ* mice were genotyped as previously published (*Arnes et al., 2012b*). Both Nkx2.2^-/- and Nkx2.2^LacZ/LacZ mice are referred to as Nkx2.2^KO mice in this paper (*Arnes et al., 2012b*). Animal maintenance and procedures were conducted in accordance with a Columbia University Institutional Animal Care and Use Committee approved protocol (AAAG3206). To minimize animal usage while allowing for statistical analysis, a minimum of three independent embryos or animals were assessed for each experiment. The *R26-tdTomato* allele was occasionally leaky. Any embryos or mice with widespread Tomato expression and/or were determined to have the recombined allele (by PCR analysis) outside of the endocrine pancreas were excluded from the study.

## Immunohistochemistry

### Fixation

Samples were fixed in 4% paraformaldehyde (PFA). E15.5 embryos were fixed for 3 hr (h) at 4°C, and P0 neonates were fixed overnight at 4°C. Samples were washed in phosphate-buffered saline (PBS), and then transferred to 30% sucrose/PBS overnight at 4°C. Samples were embedded and frozen in optimum cutting temperature (O.C.T.) and stored at −80°C. 8 μm sections were taken and stored at −80°C.

### Staining

Sections were washed in PBS and 0.1% Triton/PBS (PBST) and blocked in 2% donkey serum (DS)/PBST for 30 min (min) at room temperature (RT). The primary antibody was diluted in 2% DS/PBST and incubated on sections overnight at 4°C. Sections were washed in PBS and PBST and incubated 2–3 hr at RT with secondary antibody (Jackson Immuno Research, West Grove, PA) diluted in 2% DS/PBST at 1:500. Sections were washed in PBS and PBST and incubated with DAPI (Invitrogen TD21490) diluted in PBS at 1:1000 for 15 min at RT. Sections were washed in PBS and mounted with fluorescent mounting medium (DAKO Cat# S3023; Agilent Technologies, Santa Barbara, CA).

### Primary antibodies

Guinea pig α-Insulin (Dako Cat# A0564 RRID:AB_10013624,1:500), rabbit α-Nkx2.2 (Sigma-Aldrich, St. Louis MO; Cat# HPA003468 RRID:AB_1079490, 1:400), goat α-Ghrelin (Santa Cruz Biotechnology, Dallas TX; Cat# sc-10368 RRID:AB_2232479, 1:500), rabbit α-Glucagon (Dako Cat# A0565 RRID:AB_10013726, 1:800), rabbit α-Pancreatic Polypeptide (Innovative Research, Novi, MI; Cat# 18–0043 RRID:AB_140259, 1:200), guinea pig α-Pdx1 (Beta cell biology consortium (BCBC), 1:400), rabbit α-Amylase (Sigma-Aldrich, 1:500), rabbit α-Somatostatin (Phoenix Pharmaceuticals, Burlingame, CA; 1:200), rabbit α-Gastrin (Cell Marque, Rocklin, CA; 1:300). Rabbit α-Neurog3 (BCBC, 1:200) was used following an antigen retrieval step.

### Microscopy

Images were taken using a Zeiss Confocal LSM 710 microscope and processed with Zen, ImageJ, and Adobe Photoshop software.

### Cell quantification

For each neonate, every twentieth section was analyzed for insulin^+, pancreatic polypeptide^+, glucagon^+, somatostatin^+, ghrelin^+, and Tomato^+ cell quantification throughout the entire pancreas. Tiled images were taken at 20x objective with a Leica fluorescent microscope (DM5500/MZ16F) and processed with LAF software. Cells were counted and normalized to the total number of Tomato^+ cells or total pancreatic area measured in arbitrary units using ImageJ software. An n ≥ 3 biological replicates (individual pancreata from different animals) were used. All values are expressed as mean

± SEM. Statistical analysis was performed using two-tailed student's unpaired *t*-test since independent animals were used in the analysis. Significance was achieved with p-value <0.05.

## RNA analysis

### RNA extraction

Whole pancreata were dissected in PBS and transferred to RNA*later* (Ambion AM7021). Pancreata were incubated at 4°C overnight and stored at −20°C until RNA extraction. Pancreata were homogenized and total RNA was isolated following the Qiagen RNeasy Mini kit.

### Quantitative reverse transcription-polymerase chain reaction (qRT-PCR) analysis

A normalized starting quantity of RNA (200 ng-1 µg) isolated from whole pancreas and random hexamers were used to synthesize cDNA following the SuperScript III reverse transcriptase protocol (Invitrogen 18080–044). qRT-PCR was performed using 200 ng cDNA and master mix (Eurogentec) and analyzed using the BioRad CFX96 Real-Time System. Genes were normalized to *CyclophilinB* (probe 5': tggtacggaaggtggag, forward primer 5': gcaaagttctagagggcatgga, reverse primer 5': cccggctgtctgtcttggt) and to littermate controls. An n ≥ 3 biological replicates (individual pancreata from different animals) were used at all ages; technical replicates were also completed in triplicate during qRT-PCR plate preparation. All values are expressed as mean ± SEM. Statistical analysis was performed using two-tailed student's unpaired *t*-test since independent animals were used in the analysis. Significance was achieved with p-value <0.05.

#### Applied Biosystems TaqMan AODs

*Insulin1* (Mm01950294_s1), *Insulin2* (Mm00731595_gH), *Glucagon* (Mm00801712_m1), *Pancreatic polypeptide* (Mm00435889_m1), *Ghrelin* (Mm00445450_m1), *Somatostatin* (Mm00436671_m1), *Gastrin* (Mm00439059_g1), *ChromograninA* (Mm00514341_m1), *Arx* (Mm00545903_m1), *Pdx1* (Mm00435565_m1), *Pax6* (Mm00443081_m1), *Isl1* (Mm00517585_m1), *Sox9* (Mm00448840_m1), *Pax4* (Mm01159036_m1).

#### Applied Biosystems Primer/TaqMan Probe sets

*Nkx2.2* (probe 5': ccattgactctgccccatcgctct, forward primer 5': cctccccgagtggcagat, reverse primer 5': gagttctatcctctccaaaagttcaaa), *Neurogenin3* (probe 5': cctgcgcttcgcccacaact, forward primer 5': gacgccaaacttacaaag, reverse primer 5': gtcagtgcccagatgt), *Nkx6.1* (probe 5': tctggttccagaaccgcagga, forward primer 5': cggagagtcaggtca, reverse primer 5': tgcgtgcttctttctc), *Mafa* (probe 5': cggcgcacgctcaagaaccg, forward primer 5': catccgactgaaacagaag, reverse primer 5': ctcgctctccagaatgtgccgctgc), *Mafb* (probe 5': cgcgtccagcagaaacatcacc, forward primer 5': ccagtcgtgcaggtat, reverse primer 5': tgcgtcttctcgttctc), *Neurod1* (probe: ABI 185747095, forward primer 5': ccagcccactaccaatttgg, reverse primer 5': gggttctgctcaggcaagaa).

### RNA-sequencing (RNA-Seq)

RNA was isolated from whole pancreata at E15.5 from $Nkx2.2^{\triangle endo}$ (n = 3 biological replicates), $Nkx2.2^{KO}$ (n = 3 biological replicates), and respective littermate control embryos (n = 3 biological replicates for each mutant). Total RNA concentration and quality were measured using the Agilent Bioanalyzer RNA Nano chip. Library preparation was performed using Illumina TruSeq RNA prep kit. Libraries were then sequenced using Illumina HiSeq2000 at the Columbia Genome Center. RNA-Seq was performed with 30 million single end 100 bp reads. RTA (Illumina) was used for base calling and CASAVA (version 1.8.2) was used for converting BCL to fastq format, coupled with adaptor trimming. Reads were mapped to a reference genome (Mouse: UCSC/mm9) using Tophat (*Trapnell et al., 2009*) (version 2.0.4) with four mismatches (–read-mismatches = 4) and 10 maximum multiple hits (–max-multihits = 10). Relative abundance (expression level) of genes and splice isoforms was determined using cufflinks (*Trapnell et al., 2010*) (version 2.0.2) with default settings. Differentially expressed genes were calculated using Deseq (*Anders and Huber, 2010*). Dataset overlap was determined with a Fisher's Exact test; significance was achieved with p-value <0.05. Data is available at GSE80444: reviewer link - (http://www.ncbi.nlm.nih.gov/geo/query/acc.cgi?token=izyriugqnbmdpmp&acc=GSE80444).

## ChIP-Sequencing (ChIP-Seq) comparative analysis

Nkx2.2 (*Gutiérrez et al., 2017*) (GSE79725), Rfx6 (*Piccand et al., 2014*) (GSE62844), and Neurod1 (*Tennant et al., 2013*) (GSE30298) ChIP-Seq datasets were aligned to mm9 and visualized using IGV (*Robinson et al., 2011*; *Thorvaldsdóttir et al., 2013*). Rfx6 bound genes were defined as genes containing Rfx6 binding within 100 kb of of their transcriptional start site (TSS) (*Piccand et al., 2014*; GSE62844). Neurod1 bound genes were defined as those containing Neurod1 binding within 100 kb of TSS (*Tennant et al., 2013*) (GSE30298). Nkx2.2 bound genes were defined as those containing Nkx2.2 binding within 100 kb of TSS (*Gutiérrez et al., 2017*) (GSE79725). Nkx2.2 direct targets were determined by cross-referencing the $Nkx2.2^{\triangle endo}$ RNA-Seq data (adjusted p-value <0.05, GSE80444; http://www.ncbi.nlm.nih.gov/geo/query/acc.cgi?token=azcbauygvrsznaf&acc=GSE79785) with the Nkx2.2 bound genes (*Gutiérrez et al., 2017*). Dataset overlap was determined with a Fisher's Exact test; significance was achieved with p-value <0.05. Gene ontology (GO) analysis was performed (*Gene Ontology Consortium, 2015*).

## Acknowledgements

We thank Andrew Leiter (UMass Medical School) for the *Neurog3-Cre* mouse line. We thank members of the Sussel lab for helpful discussions throughout the project and especially Dina Balderes for assistance with mouse maintenance and tissue collection. We thank the Columbia University Genome Center for performing the RNA sequencing and assistance with the analysis. Additional core facility support was provided from the Columbia DERC (NIH P30 DK063608) and the Herbert Irving Comprehensive Cancer Center. Funding for the project was provided by NIH R01 DK082590 (LS), NIH T32GM007088 and T32DK007328 (AJC), a CONACYT grant from the Mexican government (GD) and NIH F31 DK107028 (RAS).

## Additional information

### Funding

| Funder | Grant reference number | Author |
|---|---|---|
| National Institute of Diabetes and Digestive and Kidney Diseases | R01 DK082590 | Lori Sussel |
| National Institute of General Medical Sciences | Graduate student training grant, T32GM007088 | Angela J Churchill |
| National Institute of Diabetes and Digestive and Kidney Diseases | Graduate student training grant, T32DK007328 | Angela J Churchill |
| National Institute of Diabetes and Digestive and Kidney Diseases | NRSA individual predoctoral award, F31 DK107028 | Ruth A Singer |
| Consejo Nacional de Ciencia y Tecnología | Stipend and tuition scholarship from the Mexican government | Giselle Dominguez Gutiérrez |

The funders had no role in study design, data collection and interpretation, or the decision to submit the work for publication.

### Author contributions

AJC, Performed the majority of experiments in the study, Co-wrote the mansucript with LS, Conception and design, Acquisition of data, Analysis and interpretation of data, Drafting or revising the article; GDG, Acquired ChIP-Seq data, reviewed the manuscript, Analysis and interpretation of data, Drafting or revising the article, Contributed unpublished essential data or reagents; RAS, Assisted with computational analysis of RNA-Seq and ChIP-Seq data, reviewed and edited the manuscript, Analysis and interpretation of data, Drafting or revising the article; DSL, Acquisition of data and contributed a figure to the manuscript, Acquisition of data, Drafting or revising the article; KAF, Acquisition of data, Acquisition of data; LS, Conceived of the study, worked with AJC on the planning,

implementation and interpretation of the data, Co-wrote the manuscript with AJC, Conception and design, Analysis and interpretation of data, Drafting or revising the article

**Author ORCIDs**
Lori Sussel, ⓘ http://orcid.org/0000-0003-2832-1319

**Ethics**
Animal experimentation: This study was performed in strict accordance with the recommendations in the Guide for the Care and Use of Laboratory Animals of the National Institutes of Health. All of the animals were handled according to approved institutional animal care and use committee (IACUC) protocols (AC-AAAG3206) of Columbia University. Every effort was made to minimize suffering.

## Additional files

### Major datasets

The following datasets were generated:

| Author(s) | Year | Dataset title | Dataset URL | Database, license, and accessibility information |
|---|---|---|---|---|
| Gutierrez GD, Sussel L, Tsirigos A, Kelly SM | 2016 | Genome-wide assessment of Nkx2.2 binding sites in MIN6 pancreatic beta cell line | https://www.ncbi.nlm.nih.gov/geo/query/acc.cgi?acc=GSE79785 | Publicly available at the NCBI Gene Expression Omnibus (accession no: GSE79785) |
| Churchill AJ, Sussel LG | 2016 | RNA-Seq analysis of E15.5 pancreas of both whole body Nkx2.2 mutant embryos and endocrine progenitor specific Nkx2.2 mutant embryos | https://www.ncbi.nlm.nih.gov/geo/query/acc.cgi?acc=GSE80444 | Publicly available at the NCBI Gene Expression Omnibus (accession no: GSE80444) |

The following previously published datasets were used:

| Author(s) | Year | Dataset title | Dataset URL | Database, license, and accessibility information |
|---|---|---|---|---|
| Hoffman BG, Hoodless PA, Jones SJ | 2014 | Rfx6 target genes in the Min6b1 cell line, a pancreatic beta cell model | http://www.ncbi.nlm.nih.gov/geo/query/acc.cgi?acc=GSE30298 | Publicly available at the NCBI Gene Expression Omnibus (accession no: GSE62844) |
| Hoffman BG, Hoodless PA, Jones SJ | 2013 | Identification and analysis of pancreatic islet specific enhancers | http://www.ncbi.nlm.nih.gov/geo/query/acc.cgi?acc=GSE30298 | Publicly available at the NCBI Gene Expression Omnibus (accession no: GSE30298) |

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
