## [Decision Letter]

Thank you for submitting your article "Ablation of Nkx2.2 from Ngn3+ progenitor cells redefines the transcriptional hierarchy regulating β cell specification" for consideration by *eLife*. Your article has been reviewed by three peer reviewers, one of whom is a member of our Board of Reviewing Editors, and the evaluation has been overseen by and Fiona Watt as the Senior Editor. The following individual involved in review of your submission has agreed to reveal his identity: Yuval Dor (Reviewer #3).

The reviewers have discussed the reviews with one another and the Reviewing Editor has drafted this decision to help you prepare a revised submission.

Summary:

This manuscript investigates the spatio-temporal requirements for the transcription factor Nkx2.2 in pancreatic endocrine cells. To do this, the authors ablate Nkx2.2 specifically in endocrine progenitor cells using Ngn3-Cre (called *Nkx2.2^Δendo^*), and compare the resulting phenotype with that of the global Nkx2.2 KO. They find that the endocrine-specific KO largely recapitulates the global KO phenotype in the pancreas, including severe β cell deficiency, establishing that Nkx2.2 is required to orchestrate β cell formation in progenitor cells already committed to an endocrine fate – in other words downstream to Neurog3 (Ngn3).

This is interesting as Nkx2.2 has been proposed as an early upstream regulator of Neurog3 and other key β cell transcription factors (Rfx6 and NeuroD1), based on the global KO and expression data, but Neurog3, Rfx6 and NeuroD1 expression were not affected in the Nkx2.2endo pancreas. This suggests that although Nkx2.2 is important for expression of Neurog3, Rfx6 and NeuroD1, its main function is downstream to these factors.

The authors then investigate the relationship between Nkx2.2, Rfx6 and NeuroD1 in regulation of β cell fate, through analysis of existing ChIPseq and new RNAseq data, demonstrating co-binding of 2 or more of these transcription factors at a high proportion of Nkx2.2 targets. From this, they conclude that in endocrine progenitors Nkx2.2 functions as an essential component of a modular network including Rfx6 and NeuroD1, which is required for pancreatic β cell fate.

Overall this is a straightforward genetic analysis that has led to a surprising and interesting finding on the wiring of transcriptional regulators of β cell development, placing the key function of Nkx2.2 much later than previously thought.

Essential revisions:

1) Please include in Figure 1 quantitative data showing that the area of tomato expression is equivalent in the control and Nkx2.2endo pancreas. Similarly, please provide quantitative analysis of the number of cells of the various sub-lineages in the global versus conditional knockout.

2) Please ensure that caveats related to the ChIP-seq datasets used (which are from insulinoma cell lines (MIN6)) are fully discussed. In particular, we would encourage inclusion of a discussion noting that insulinoma cell lines in vitro may have limited comparability to embryonic proto-endocrine cells. Similarly, discussion would be welcome of the limitations arising when population-derived datasets are compared, where the phenotype under consideration affects the precursors of some but not all sub-lineages in the population under study.

3) Please review the manuscript for clarity. In particular, the reviewers have requested a more lucid presentation of what pathway Nkx2.2 is actually affecting – i.e. to what extent is it required in endocrine progenitors for production of all endocrine lineages, rather than just for the β cell lineage as sometimes implied in the current text. Similarly, inclusion of more details of normal gene expression patterns in the relevant sections of text is encouraged, for instance, for Rfx6, Neurod1, etc., are these genes all expressed in proto-β-cells alone, or is there evidence for broader expression? We also request that the Abstract is rewritten for clarity.

4) Please also attend to the following points:

a) The paper suggests a role for Nkx2.2 in endocrine progenitors. However Ngn3+ progenitor cells are short lived before they give rise to hormone producing cells. Is it possible that by the time that Nkx2.2 protein is eliminated, the affected cells are already young β or α cells rather than Ngn3+ progenitors? This possibility does not detract from the importance of the story, but the distinction is important because of the fundamental difference between affecting fate of progenitors and directing terminal differentiation of β cells. Specifically, please indicate whether Nkx2.2-deficient, tomato+ cells that are Ngn3+ are present in the *Nkx2.2^Δendo^* mice. If not, it is hard to argue that the deletion affects progenitor cells and the text should be adjusted accordingly.

b) The conclusions about cell fate and identity should be clarified, preferably also in the model (Figure 5). What happens to progenitor cells that lose Nkx2.2? The authors show that Ngn3 expression is reduced as in wild type, and that the cells do express general endocrine markers. What is less clear is whether lack of Nkx2.2 causes a shift in fate to alternative endocrine cell types like epsilon and pp, causes the formation of aberrant endocrine cells with mixed or no hormone expression, or does allow β cell specification (as reflected by expression of NeuroD1 for example) but not terminal differentiation. The transcriptome of the tomato+ cells could potentially be used to address this question, although caution is needed regarding the possibility raised in the point made above, that deletion in this model occurs after commitment to a β cell fate.

c) The authors are encouraged to speculate about the role of Nkx2.2 in early pancreatic progenitors given the new findings – e.g. could it have a key function at this stage that is masked by redundancy?

---

## [Author Response]

*Essential revisions:*

*1) Please include in Figure 1 quantitative data showing that the area of tomato expression is equivalent in the control and Nkx2.2endo pancreas. Similarly, please provide quantitative analysis of the number of cells of the various sub-lineages in the global versus conditional knockout.*

We have added the quantitative data to Figure 1—figure supplement 1. We counted the number of tomato labeled cells as a function of pancreas area. There is no difference between wild type and mutant. We have also provided the quantitative analysis of the numbers of sub-lineage islet cells in the *Nkx2.2^Δendo^* mice with the corresponding images presented in Figure 2—figure supplement 1. The cell number changes are consistent with the RNA changes. They are not as severe as the global KO, likely due to Cre efficiency – which we discuss in the manuscript.

*2) Please ensure that caveats related to the ChIP-seq datasets used (which are from insulinoma cell lines (MIN6)) are fully discussed. In particular, we would encourage inclusion of a discussion noting that insulinoma cell lines* in vitro *may have limited comparability to embryonic proto-endocrine cells. Similarly, discussion would be welcome of the limitations arising when population-derived datasets are compared, where the phenotype under consideration affects the precursors of some but not all sub-lineages in the population under study.*

We have included discussion of these caveats. It is important to note that we compare all ChIP-seq data with in vivo expression data to ensure a gene is a bona fide target. We also agree that there are caveats associated with population-derived datasets and we have added this discussion to paragraph three of the Discussion section.

*3) Please review the manuscript for clarity. In particular, the reviewers have requested a more lucid presentation of what pathway Nkx2.2 is actually affecting – i.e. to what extent is it required in endocrine progenitors for production of all endocrine lineages, rather than just for the β cell lineage as sometimes implied in the current text. Similarly, inclusion of more details of normal gene expression patterns in the relevant sections of text is encouraged, for instance, for Rfx6, Neurod1, etc., are these genes all expressed in proto-β-cells alone, or is there evidence for broader expression? We also request that the Abstract is rewritten for clarity.*

We had deliberately focused on the β cell lineage in our initial study, but we have now added the additional information about the other endocrine lineages. Much of the requested information was already provided in the Introduction, although we have added additional information about the pathways Nkx2.2 is affecting to the Introduction and Discussion. We have also rewritten the Abstract to improve clarity, but maintained the 150-word limit.

*4) Please also attend to the following points:a) The paper suggests a role for Nkx2.2 in endocrine progenitors. However Ngn3+ progenitor cells are short lived before they give rise to hormone producing cells. Is it possible that by the time that Nkx2.2 protein is eliminated, the affected cells are already young β or α cells rather than Ngn3+ progenitors? This possibility does not detract from the importance of the story, but the distinction is important because of the fundamental difference between affecting fate of progenitors and directing terminal differentiation of β cells. Specifically, please indicate whether Nkx2.2-deficient, tomato+ cells that are Ngn3+ are present in the Nkx2.2^Δendo^mice. If not, it is hard to argue that the deletion affects progenitor cells and the text should be adjusted accordingly.*

This is a valid point, but was also a difficult experiment since the Ngn3 and Nkx2.2 antibodies are both made in rabbit and the conditions we need to use for these antibodies quench the tomato signal. However, we have added a figure to Figure 1—figure supplement 1 that shows low and high power images of adjacent sections of a pancreas from an e15.5 mouse. This data clearly shows that the majority of tomato-labeled positive cells are Nkx2.2-deficient and Ngn3 positive. We have added this statement to the text as well since it supports our interpretation and conclusions.

*b) The conclusions about cell fate and identity should be clarified, preferably also in the model (Figure 5). What happens to progenitor cells that lose Nkx2.2? The authors show that Ngn3 expression is reduced as in wild type, and that the cells do express general endocrine markers. What is less clear is whether lack of Nkx2.2 causes a shift in fate to alternative endocrine cell types like epsilon and pp, causes the formation of aberrant endocrine cells with mixed or no hormone expression, or does allow β cell specification (as reflected by expression of NeuroD1 for example) but not terminal differentiation. The transcriptome of the tomato+ cells could potentially be used to address this question, although caution is needed regarding the possibility raised in the point made above, that deletion in this model occurs after commitment to a β cell fate.*

The reviewers raise an important point that we have clarified in the text and in the model figure. We have shown previously using genetic lineage tracing that there is a shift in fate when Nkx2.2 is lost during pancreas development (using a null allele). We are unable to lineage label the β cells in the *Nkx2.2^Δendo^* mice since these mice already have Ngn3:Cre, which would confound any lineage labeling experiments with another Cre. However, we do not observe insulin producing cells expressing other hormones, suggesting we are not generating cells of mixed fate. We have clarified this point in the text. Furthermore, we have now determined in a separate study that deletion of Nkx2.2 in the β cells (after their commitment to the β cell fate) has a different phenotype than the *Nkx2.2^Δendo^* mice. This manuscript is now in press and we discuss these differences in this paper in the Discussion section.

*c) The authors are encouraged to speculate about the role of Nkx2.2 in early pancreatic progenitors given the new findings – e.g. could it have a key function at this stage that is masked by redundancy?*

We have extended this discussion.